# Diet Impacts on Gene Expression in Healthy Colon Tissue: Insights from the BarcUVa-Seq Study

**DOI:** 10.3390/nu16183131

**Published:** 2024-09-16

**Authors:** Mireia Obón-Santacana, Ferran Moratalla-Navarro, Elisabet Guinó, Robert Carreras-Torres, Virginia Díez-Obrero, David Bars-Cortina, Gemma Ibáñez-Sanz, Lorena Rodríguez-Alonso, Alfredo Mata, Ana García-Rodríguez, Matthew Devall, Graham Casey, Li Li, Victor Moreno

**Affiliations:** 1Unit of Biomarkers and Susceptibility (UBS), Oncology Data Analytics Program (ODAP), Catalan Institute of Oncology (ICO), 08908 L’Hospitalet del Llobregat, Barcelona, Spain; 2ONCOBELL Program, Bellvitge Biomedical Research Institute (IDIBELL), 08908 L’Hospitalet de Llobregat, Barcelona, Spain; 3Consortium for Biomedical Research in Epidemiology and Public Health (CIBERESP), 28029 Madrid, Spain; 4Department of Clinical Sciences, Faculty of Medicine, University of Barcelona, 08907 L’Hospitalet de Llobregat, Barcelona, Spain; 5Digestive Diseases and Microbiota Group, Department of Gastroenterology, Girona Biomedical Research Institute (IDIBGI), Hospital Universitari de Girona Dr. Josep Trueta, 17190 Salt, Girona, Spain; 6Gastroenterology Department, Bellvitge University Hospital, 08907 L’Hospitalet de Llobregat, Barcelona, Spain; 7Digestive System Service, Moisés Broggi Hospital, 08970 Sant Joan Despí, Spain; 8Endoscopy Unit, Digestive System Service, Viladecans Hospital-IDIBELL, 08840 Viladecans, Barcelona, Spain; 9Department of Family Medicine, University of Virginia, Charlottesville, VA 22903, USA; 10Department of Genome Sciences, University of Virginia, Charlottesville, VA 22903, USA

**Keywords:** diet, dietary patterns, gene expression, tissue, colon, nutrigenomics

## Abstract

(1) Introduction: The global rise of gastrointestinal diseases, including colorectal cancer and inflammatory bowel diseases, highlights the need to understand their causes. Diet is a common risk factor and a crucial regulator of gene expression, with alterations observed in both conditions. This study aims to elucidate the specific biological mechanisms through which diet influences the risk of bowel diseases. (2) Methods: We analyzed data from 436 participants from the BarcUVa-Seq population-based cross-sectional study utilizing gene expression profiles (RNA-Seq) from frozen colonic mucosal biopsies and dietary information from a semi-quantitative food frequency questionnaire. Dietary variables were evaluated based on two dietary patterns and as individual variables. Differential expression gene (DEG) analysis was performed for each dietary factor using edgeR. Protein–protein interaction (PPI) analysis was conducted with STRINGdb v11 for food groups with more than 10 statistically significant DEGs, followed by Reactome-based enrichment analysis for the resulting networks. (3) Results: Our findings reveal that food intake, specifically the consumption of blue fish, alcohol, and potatoes, significantly influences gene expression in the colon of individuals without tumor pathology, particularly in pathways related to DNA repair, immune system function, and protein glycosylation. (4) Discussion: These results demonstrate how these dietary components may influence human metabolic processes and affect the risk of bowel diseases.

## 1. Introduction

The prevalence of gastrointestinal diseases, such as colorectal cancer and inflammatory bowel disease, is increasing worldwide [1,2]. Despite having different clinical manifestations and severities, these diseases share several risk factors, particularly those related to lifestyle, including obesity, tobacco smoking, and sedentary behavior [3,4,5]. In addition to these findings, The World Cancer Research Fund/American Institute for Cancer Research reported strong evidence that frequent dietary consumption of whole grains, fiber, dairy products, and calcium supplements decreases colorectal cancer risk, whereas processed and red meat and alcohol consumption increases risk [6]. Similarly, epidemiological studies show that a diet high in sugar, fat, and red meat can elevate the risk of inflammatory bowel disease, while a diet rich in fruits, vegetables, and fish can lower the risk [7].

Nevertheless, the study of dietary patterns has gained relevance in nutritional epidemiology, complementing the study of single components. This methodology might reflect the effect of overall diet on disease risk. For instance, patterns associated with a higher intake of plant-based foods with regular intake of fish and seafood (such as the Mediterranean diet (MedDiet) or the prudent pattern) are associated with a decreased colorectal cancer risk, while those characterized by a higher intake of red or processed meats, sweets, and high-saturated-fat products (such as the Western pattern) are associated with an increased risk [8]. Similarly, a high-quality-diet index is linked to a reduced risk of inflammatory bowel disease [9].

Although dietary patterns and some single components have been associated with different circulating biomarkers [10,11], the biological mechanisms underlying their connection to gastrointestinal diseases remain unclear and need further exploration. Nutrigenomic studies can help fill this gap. Diet is an important regulator of gene expression, and expression alterations have been reported in colorectal cancer and inflammatory bowel disease [12,13,14,15]. Peroxisome proliferator-activated receptors (PPARs) are a group of nuclear receptor proteins acting as transcription factors that regulate the expression of genes participating in cell differentiation and lipid metabolism. Dietary fatty acids, mainly mono- and polyunsaturated, directly bind PPARs and modulate their activity [16]. Likewise, previous studies that evaluated the association between dietary patterns and gene expression in peripheral blood mononuclear cells (PBMCs) have revealed different gene expression profiles according to the studied patterns [17,18,19].

Furthermore, diet plays a central role in modulating inflammation, oxidative stress, and gut microbiota (dysbiosis), all of which are linked to gastrointestinal diseases. Pro-inflammatory dietary components, such as high saturated fats, refined sugars, and processed foods, may increase the production of pro-inflammatory cytokines, exacerbating chronic inflammation [20,21]. On the other hand, diets rich in antioxidants, such as fruits and vegetables, can mitigate oxidative stress by neutralizing free radicals, thereby protecting the gut lining from oxidative damage [22,23].

Additionally, dietary components directly influence the composition of the gut microbiota. A diet high in fiber, for instance, promotes the growth of beneficial bacteria that produce short-chain fatty acids (SCFAs), which are known to support gut barrier function and reduce inflammation [24,25,26].

To the best of our knowledge, the effect of diet on gene expression profile from colon tissue has only been evaluated in a few small studies; however, they could identify several dietary components associated with gene expression [27,28,29]. The aim of the present study was to assess and replicate, in a population of healthy Spanish adults, the associations between dietary components and patterns and gene expression in colon tissue and, therefore, elucidate the biological mechanisms by which diet may influence bowel diseases.

## 2. Materials and Methods

### 2.1. Study Population and Design

BarcUVa-Seq is a population-based, cross-sectional study of 485 participants who were part of two epidemiological studies: Colobank and COLSCREEN. These studies were designed to recruit participants from the ongoing CRC screening program conducted by the Catalan Institute of Oncology at different timepoints but under similar protocols. Colobank recruited participants from 2011 to 2016 and COLSCREEN from 2016 to 2020 [30,31]. The design of the CRC screening program, based on the fecal occult blood test (FOBT), has been published elsewhere [32,33]. Briefly, all residents in the screening area (southwest of Barcelona, Catalonia, Spain) aged 50–69 are biennially invited to participate using the fecal immunochemical test (FIT, OC-Sensor, Eiken Chemical Co., Tokyo, Japan). Participants with a positive FIT result (≥20 g Hb/g feces) are referred for colonoscopy examination. According to the colonoscopy results, participants are classified following the European Guidelines for Quality Assurance in CRC [34]: (a) normal/no adenomatous pathology; (b) hyperplastic polyps; (c) low-risk adenomas; (d) intermediate-risk adenomas; (e) high-risk adenomas; or (f) CRC. Potential CRC screening program participants are excluded if they have a personal history of CRC, high-risk familial history of CRC, adenomas or inflammatory bowel disease, gastrointestinal symptoms, colonoscopy in the previous 5 years or a FIT within the last two, terminal disease, or severe disabling conditions. Additionally, both Colobank and COLSCREEN studies included participants who received an indication for colonoscopy by direct referral by their endoscopic unit.

All participants who agreed to take part in the study (Colobank or COLSCREEN) provided informed consent. For each participant, colonic mucosa biopsy samples were obtained during colonoscopy. Moreover, participants donated a blood sample and a fecal sample (only for COLSCREEN participants) and completed an extensive epidemiologic and dietary questionnaire. The BarcUVA-Seq study only includes data from endoscopically healthy colon participants (normal/no adenomatous pathology or polyp colonoscopy result) [35]. Further information about the BarcUVa-Seq project can be found at https://barcuvaseq.org/.

The CRC screening program follows the public health laws and the Organic Law on Data Protection. All procedures performed in the study involving data from human participants were in accordance with the ethical standards of the institutional research committee and with the 1964 Helsinki Declaration and its later amendments or comparable ethical standards. The Colobank and COLSCREEN protocol studies were approved by the Bellvitge University Hospital Ethics Committee (PR073/11 and PR084/16, respectively).

### 2.2. Dietary and Lifestyle Data Collection

A self-administrated semi-quantitative food frequency questionnaire (FFQ), adapted from a Spanish-validated FFQ [36], was used to assess dietary information at recruitment. The FFQ collected information with the timeframe referring to the preceding year and included portion sizes of 10 categories to assess dietary information: (1) eggs, meat, fish, shell-fish, and processed and pre-cooked meat foods; (2) vegetables and legumes; (3) fruits and nuts; (4) dairy foods; (5) cereals, bread, and starches; (6) soup, sauces, and seasonings; (7) olive oil, other oils, and fats; (8) sweets, baked goods, and miscellaneous; (9) alcoholic and carbonated beverages; and (10) non-carbonated beverages and coffee. A total of 140 food items were obtained and converted into total energy intake, macronutrients, and micronutrients using the Spanish food composition tables and additional specific sources [37].

Participants also answered an extended lifestyle and socio-demographic questionnaire about current occupation, educational level, tobacco smoking, physical activity, medical history, and drug use. Anthropometric measures (height and weight) were self-reported by Colobank participants, whereas for COLSCREEN participants, they were measured by trained personnel.

### 2.3. RNA Processing and Quality Control

Molecular profiling was previously described by Díez-Obrero et al., 2021 [35]. Briefly, RNA extraction, quantification, and sequencing of colon samples were performed using Illumina HiSeq 2500 or NovaSeq 6000 instruments. Quality control (QC) of sequences was performed by removing genes smaller than 300 bp or with less than 1 CPM in at most 100 samples. Filtered gene expression data (~15,000 genes) were normalized using the trimmed mean of M values (TMM) method. Principal component analysis (PCA) was performed to inspect for possible outliers and biases of a common nature, such as age, gender, and sequencing batch effects. Of the initial 485 participants, 25 had to be excluded as they did not pass the above QC, 21 due to lack of dietary information, and 3 due to reporting a consumption of total energy outside the predefined thresholds (men: <800 kcal or >4000 kcal/day; women: <500 kcal or >3500 kcal/day) [38], leaving 436 subjects included in the final analyses.

### 2.4. Dietary Variable and Pattern Assessment

The adherence to the MedDiet was assessed using two approaches: the Mediterranean Diet Score (MDS) [39] and the *adapted-relative* Mediterranean Diet Score (arMDS) [40]. The MDS designed by Trichopoulou et al., 2013 [39] includes 9 components with a sex-specific median as a cutoff. Participants below the median receive a value of 0 and the others a value of 1. The MDS scale ranges from 0 (representing the lowest adherence) to 9 (indicating the highest adherence) to the MedDiet. Buckland et al., 2013 [40] introduced arMDS, building upon the MDS but with 8 components, excluding alcoholic beverages. Participants are then grouped into tertiles based on their scores. To evaluate adherence to the MedDiet, foods that align with its principles are assigned scores ranging from 0 to 2, whereas foods that deviate from the diet receive reversed scoring. As a result, the arMDS scale varies from 0 (indicating the lowest adherence) to 16 (reflecting the highest adherence to the MedDiet).

Eighteen dietary variables (estimated as food groups) were energy-adjusted using the residual method (RM) described by Willet et al., 2012 [38] and classified into sex-specific quartiles based on the distribution in the full BarcUVA-Seq dataset: blue fish, white fish, seafood, red meat, white meat, cured and processed meat, dessert pastries, eggs, fruits, legumes, nuts, vegetables, cereals (rice, pasta, bread, and breakfast cereals), potatoes (cooked and fried), dairy products (milk, yogurt, dairy desserts, and cheese), coffee, olive oil, and sugar. Those food groups that were infrequently consumed by the majority of participants (such as caloric beverages and alcoholic beverages) were divided into three categories: non-consumers, sex-specific below-median values (low consumers), and sex-specific above-median values (high consumers).

Baseline dietary and anthropometric characteristics were examined using means and standard deviations (SD) for continuous variables and percentages for categorical variables by sex. Differences between male and female participants were assessed using the Wilcoxon rank-sum test for continuous variables and the Pearson *Χ*^2^ test for categorical variables. Spearman’s correlation tests were performed to evaluate the correlation between dietary variables in the dataset.

### 2.5. Differential Expression Analysis

Differential expression gene (DEG) analysis was performed for each dietary pattern and variable as a trend (increment by quartile/category) with *edgeR* [41] library package from R [42] Specifically, generalized linear models with a robust quasi-likelihood method were used to have a robust type I error control rate. Models were adjusted by age (years), sex, RNA sequencing batch (1–4), energy intake (kcal/day), and tissue location from the biopsy (ascending, transverse, or descending). The Benjamini and Hochberg false discovery rate (FDR) was set to 0.05 to filter statistically significant DEGs. As a result, one unit of log fold change (logFC) represents a two-fold variation of expression between one group and the following one.

For each dietary pattern, we also assessed which genes presented statistically significant differential expression. The adherence to the MedDiet, assessed using the MDS and the arMED, was evaluated as high adherence vs. low adherence.

Sensitivity DEG analyses were conducted on those food groups that exhibited high correlations Pearson coefficients, refining the generalized linear models by accounting for the highly correlated variable. In addition, a secondary sensitivity analysis was performed to account for the impact of BMI in the analysis. We accounted for the differences in each food group before and after adjusting the models for BMI by computing Spearman’s correlation values.

### 2.6. Functional Analysis of Food Group Expression Profiles

Protein–protein interaction (PPI) analysis was performed using the *STRINGdb* R package [43]. Briefly, human PPIs were extracted from STRING (release v11) and filtered to include only those with a combined score of 0.7 or higher, ensuring high-confidence interactions. PPI subnetworks were constructed for food groups with more than 10 significant DEGs, ensuring that each subnetwork comprised only proteins coded by genes present in DEGs of each food group analysis. Network clusters were identified using the fast greedy modularity optimization algorithm from the *igraph* R package [44]. The *ReactomePA* R package [45] was used to conduct enrichment analyses both generally and for each identified cluster.

## 3. Results

### 3.1. Study Population

Detailed descriptive characteristics of the study population are shown in Table 1. Briefly, the study comprised 436 participants (63.7% of females) with a mean age of 59.8 years. The overall mean BMI was 27.5 kg/m^2^, and there were no significant differences observed between sexes (*p*-value: 0.13). Mucosa biopsy samples were obtained along the ascending (31.9%), transverse (30.9%), and descending (37.2%) colon. Men, compared to women, had higher daily energy values as well as increased consumption of alcoholic beverages and blue fish (all *p*-values < 0.004).

Among the strongest correlations, we observed that caloric beverage intake inversely correlated with olive oil intake (Pearson correlation coefficient of −0.69), while blue fish and white fish intakes were positively correlated (0.61). Potato intake was also correlated with olive oil intake (0.58) and with caloric beverage intake (−0.50) (Figure 1). These correlations were maintained when analysis was performed by sex; however, the correlation between white and blue fish intakes was stronger in women than men (Appendix A).

### 3.2. Differential Expression Analysis

All of the included dietary factors are shown in Appendix A. In this study, we have focused on presenting the results of food groups that demonstrated the highest number of DEGs meeting the significance threshold of FDR < 0.05 (Table 2). These food groups, namely potatoes, caloric beverages, olive oil, blue fish, and alcoholic beverages, collectively contributed to a total cumulative count of 98% of DEGs at FDR < 0.05 (Appendix A). For instance, a total of 2586 genes were found to be differentially expressed in response to potato intake, with 1242 genes upregulated and 1344 genes downregulated (Table 2). The number of DEGs and percentages of upregulated genes for the other top food variables were 1786 (61% upregulated) for caloric beverages, 987 (63%) for olive oil, 538 (40%) for blue fish, and 402 (46%) for alcoholic beverages (Table 2). Seafood, cured and processed meat, cereals, and the two dietary patterns (MDS and arMDS) did not exceed the FDR threshold of 0.05 (Appendix A).

The results of sensitivity analysis for correlated variables revealed only 226 DEGs associated with potato intake (63% upregulated) (Table 2). When olive oil intake and caloric beverage intake were adjusted for potato consumption, the DEGs dropped under the threshold of 10 for functional analyses (Table 2). Therefore, based on the sensitivity analyses, the main food groups with the highest number of DEGs were blue fish, alcoholic beverages, and potatoes, accounting for 90% of the total DEGs (Appendix A). These three food groups have been examined in detail in the subsequent analyses. Results of the BMI sensitivity analysis showed slightly lower DEGs after adjusting for BMI, with logFC Spearman correlation values between food groups higher than 0.95 in all cases; therefore, no additional adjustment for BMI was needed. The results of BMI sensitivity analysis for the top five food groups are shown in Appendix A).

If we examine the logFC values in the genes common between food groups, an inverted pattern becomes evident (Appendix A). The same genes that were found downregulated when evaluating the variable blue fish were upregulated with the variable alcohol, and vice versa (number of downregulated and upregulated DEGs: 29 and 26, respectively). The DEGs in the analysis of the variable potatoes followed the same direction as those of the blue fish variable (number of downregulated and upregulated DEGs: 25 and 37, respectively). The same was observed for the genes in the variables of alcoholic beverages and caloric beverages: *HIST1H4I* and *ROMO1* were upregulated, and *INTS3* was downregulated.

### 3.3. Functional Analysis

To gain insight into the functional relations and pathways involving the DEGs, we performed Reactome enrichment analysis based on PPI networks. Table 3 and Figure 2 show the most significant pathway enrichment results for the three food groups (blue fish, alcoholic beverages, and potatoes). The specific outcomes related to these food groups can be found in Appendix A, respectively, providing a detailed account of the results. In addition, we repeated the analysis in each network cluster for the three mentioned food groups to assign the most significant pathway within each cluster. The results are shown in Appendix A. In the investigation of blue fish consumption and based on the Reactome hierarchy, we identified nine main pathways associated with this food group. Specifically, two pathways were classified under the “Cell Cycle” category, two under the “Membrane Trafficking” category, and the remaining five under the “Cellular Response to Stimuli”, “Metabolism of RNA”, “Diseases of Signal Transduction”, “Adaptive Immune System”, and “DNA Repair” categories. Within these pathways, we examined a total of 260 genes, of which 30 were found to be related to the “Cell Cycle Checkpoint” pathway, and all of the DEGs were downregulated. Notably, several of these genes, including *BRCA1*, *RMI1*, *BARD1*, and *RAD50*, were also found to be associated with the “Resolution of D-loop Structures through Holliday Junction Intermediates” pathway, categorized under the “DNA Repair” category, which also showed downregulated tendencies.

When we analyzed the effects of alcoholic beverages, our analyses revealed four main pathways categorized as “Immune System”, “Metabolism”, “Metabolism of Proteins”, and “Membrane Trafficking”. None of the resulting genes were shared among these pathways. All seven genes identified in the first pathway were found to be downregulated, and all of the genes identified in the metabolism pathways, related to oxidative stress, were upregulated, while six out of eleven genes found in the last pathway were downregulated.

Potatoes consumption also yielded significant results in several biological pathways. Among these, five merged as the most notable, encompassing “RNA metabolism”, “Immune system regulation”, “Cellular response to stimuli, “Asparagine *N*-linked Glycosylation”, and “SUMO E3 Ligases SUMOylate Target Proteins”. Particularly, the last two pathways belong to the same category of “Metabolism of Proteins”; however, none of the genes observed were common between them. Primarily, most of the identified pathways contained genes that were found to be upregulated. However, an exception was observed in the “SUMO E3 Ligases SUMOylate Target Proteins” pathway, where five out of the seven DEGs were downregulated.

## 4. Discussion

In this study, we have presented evidence that dietary parameters influence gene expression levels in colonic mucosal tissue among individuals without tumor pathology. Specifically, the variables that provided the most significant evidence were blue fish, alcoholic beverages, and potatoes.

Blue fish is the main source of ω-3 polyunsaturated fatty acids (PUFAs), particularly eicosapentaenoic acid (EPA, 20:5ω-3) and docosahexaenoic acid (DHA, 22:6ω-3). The human body is capable of synthesizing only small amounts of EPA and DHA from alpha-linolenic acid, making it essential to have a proper intake of these. The health benefits of ω-3 intake are widely known. In addition to being integral to cell membranes, enhancing their permeability and elasticity, ω-3 fatty acids are the base for producing *n*-3 eicosanoids, including prostaglandins, thromboxanes, leukotrienes, and neuroprotectins. These molecules are involved in physiological processes such as inflammation, blood clotting, immune response, cell proliferation and differentiation, and the regulation of blood vessel dilation and constriction [46,47]. Several epidemiological studies have observed an association between PUFAs and the levels of inflammatory markers in blood [48,49]. Nevertheless, none of the DEA genes identified in the current study were specifically linked with inflammatory pathways, but we did observe associations related to the cell cycle, disease signaling, immune system, and cellular responses to heat stress, with this last pathway also being linked to cell cycle regulation and immune response [50]. Although these associations have not been reported in any other study using frozen colonic human tissue samples, there is evidence supporting our findings. For instance, certain authors have observed that ω-3 intake inhibited cell proliferation and growth in human colon cancer [51] and cell proliferation in embryonic stem cells in mice [52] and also downregulated genes related to cell proliferation, cell cycle, and the regulation of transcription in rat cardiomyocytes [53]. However, it is important to acknowledge some literature discrepancies, as previous research has shown that the consumption of ω-3 supplements can induce the expression of genes related to cell function in peripheral blood mononuclear cells [54]. This variability in the results may indicate, among others, the presence of complex unknown mechanisms related to how fatty acids from blue fish interact depending on the biological matrix studied [55].

It is well established that alcohol consumption increases oxidative stress and decreases the antioxidant capacity of the cell [56,57]. This occurs because alcohol metabolism, primarily mediated by alcohol dehydrogenase and aldehyde dehydrogenases at low alcohol concentrations and by catalase and cytochrome P450s (*CYP2E1*) at high concentrations, results in the generation of derivate compounds, including reactive oxygen species (ROS), and decreases the levels of glutathione [58]. These ROS can induce damage in tissue and cells [59]. These findings are in complete agreement with our observations, where we found that all genes associated with the citric acid metabolic pathways, electron transport chain, and mitochondrial translation capacity were overexpressed. This outcome strongly underscores the significance of reducing or even completely abstaining from alcohol consumption in alignment with international public health guidelines [60].

Potatoes are an important source of carbohydrates in many cultures and are typically classified as a nutrient-dense food that is rich in starch, provides a modest amount of protein, and has minimal fat content. Due to their high starch content, potatoes have a high glycemic index, which can vary depending on the variety, cooking method, and preparation. In terms of micronutrients, potatoes are an excellent source of vitamin C, B vitamins (such as B6 and folate), potassium, magnesium, and iron, among others. Additionally, potatoes contain various polyphenols, which have antioxidant properties, but the specific classes of polyphenols can vary depending on the potato variety [61,62]. Therefore, it is understandable that the metabolic pathway of the immune regulation system was overexpressed in our study, as all of these micronutrients are linked to immune system function [63]. Further, our results show two metabolic pathways classified under the category “Metabolism of proteins” in which potatoes might be involved. Specifically, we found the glycosylation pathway. Protein glycosylation, which entails adding carbohydrate molecules to proteins, is one of the most common modifications that occur during and after protein synthesis. This process facilitates a variety of essential functions crucial for tissue development and maintaining the balance of mucosal surfaces, including those in the gastrointestinal tract. These functions can be modulated by a combination of host and environmental factors, such as diet [64,65]. Changes in the expression of intestinal epithelial glycans have been linked with colorectal cancer, ulcerative colitis, and Crohn’s disease [66,67]; however, none of the genes identified in this pathway have been identified as genes associated with the risk of developing inflammatory bowel disease or colorectal cancer in the current study. Based on the information we currently have, there is no reason to reduce potato consumption. At present, there is no international consensus on a recommended potato intake. However, it is commonly advised that potatoes are best consumed cooked or steamed, as they are typically prepared in traditional dishes, but fried potatoes and chips should be avoided [68].

The results presented in this cross-sectional analysis, which has the largest sample size to date, are consistent with previously published findings, providing evidence that dietary intake affects gene expression in the colon [27,28,29]. However, we were unable to replicate any specific observed outcomes, and direct comparisons are challenging due to significant differences in study parameters. For instance, unlike other published studies where RNA-Seq data were derived from normal colonic mucosa in colon cancer cases (formalin-fixed paraffine-embedded tissue), BarcUVa-Seq samples were obtained from superficial mucosal biopsies in individuals undergoing colonoscopy without any tumor pathology, providing an optimal representation of normal colonic epithelial physiology. Additionally, previous studies gathered dietary information using a diet history questionnaire with food units measured in standard servings per day. In contrast, our participants completed a food frequency questionnaire with food units measured in grams or micrograms per day. Finally, it should be noted that previous studies, including the one we currently present, have not been able to account for the cooking method for all variables, which may affect gene expression. More studies with this level of detail are required to obtain more accurate results. One of the strengths of the present study is that intake variables have been adjusted using the residual method, which accounts for variations in total energy intake among individuals with differing energy needs. This adjustment aims to decrease bias and improve accuracy when studying associations between specific food items and health outcomes [38]. Likewise, we have been stringent in setting the FDR threshold, aiming for only a 5% false discovery rate in our study, thereby achieving a reasonable level of confidence in positive results. Interestingly, neither of the two dietary patterns studied (MDS and arMDS) exceeded the strict FDR threshold of 0.05 and, therefore, were not evaluated further in subsequent analyses. This could be attributed to subtle differences among categories. In previous studies using the same threshold of 0.05 in a sensitivity analysis, only one gene, *TXNDC17*, was observed to be upregulated with a prudent dietary pattern (high in fruits, vegetables, fish, and whole grains), which also showed upregulation with vegetable intake [29]. However, we did not observe this clear association with our MDS (upregulated; FDR: 0.27) or with the intake of fruits or vegetables, where this gene had a logFC value near the null.

## 5. Conclusions

Overall, the present study provides comprehensive evidence that food intake impacts gene expression in the colon of individuals who do not have tumor pathology. Specifically, blue fish intake, alcohol consumption, and potato intake were associated with regulation (up and/or down) of genes classified in pathways such as DNA repair, immune system function, and the glycosylation of proteins, among others. These findings illustrate how these dietary components may influence human metabolic processes and impact bowel disease risk.

## Figures and Tables

**Figure 1 nutrients-16-03131-f001:**
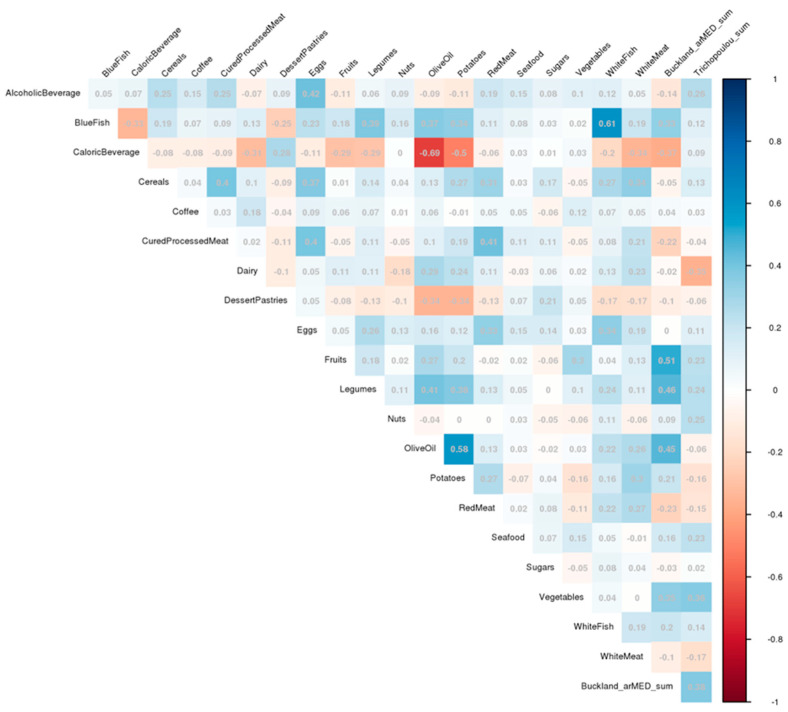
Correlation plot of dietary patterns and food group variables adjusted by the residual method.

**Figure 2 nutrients-16-03131-f002:**
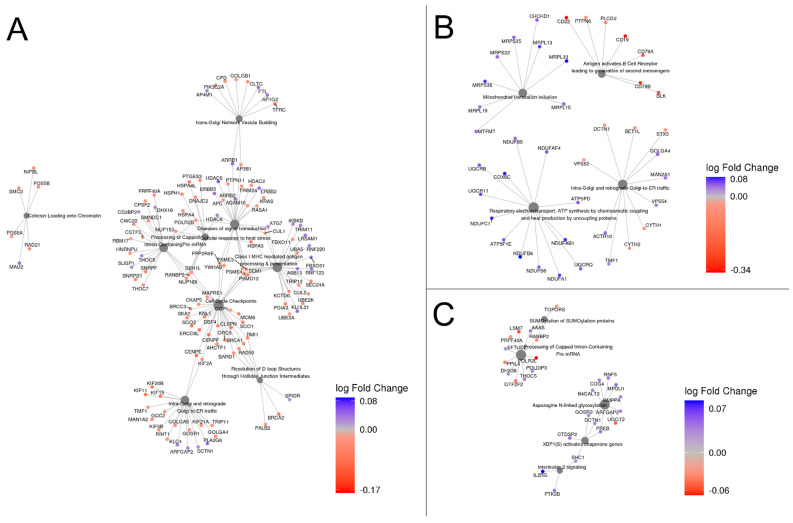
Network plot of top enriched pathways and their genes involved for the top three food groups. (**A**) Blue fish, (**B**) alcoholic beverages, and (**C**) potatoes. Grey nodes represent pathways, with size proportional to the number of DEGs within each pathway. Genes are color-coded based on their expression levels: deep red indicates genes that are upregulated in low consumers, while deep blue indicates genes that are upregulated in high consumers. The gradient from red to blue represents varying levels of log fold change values.

**Table 1 nutrients-16-03131-t001:** Descriptive statistics of the present study.

	All Participants	Female	Male	
	Mean (sd)/N (%)	Mean (sd)/N (%)	Mean (sd)/N (%)	*p*-Value *
Sex		277 (63.5)	159 (36.5)	
Age	59.8 (7.0)	60.0 (7.1)	59.5 (6.7)	0.35
Location				
Ascending	139 (31.9)	88 (63.3)	51 (36.7)	0.97
Transverse	135 (30.9)	85 (63.0)	50 (37.0)	0.97
Descending	162 (37.2)	104 (64.2)	58 (35.8)	
Batch				
1	196 (45.0)	126 (64.3)	70 (35.7)	0.90
2	94 (21.6)	61 (64.9)	33 (35.1)	0.90 0.13
3	46 (10.6)	27 (58.7)	19 (41.3)	
4	100 (22.9)	63 (63.0)	37 (37.0)	
BMI (kg/m^2^)	27.5 (4.2)	27.3 (4.6)	27.7 (3.6)	
Energy intake (kcal/day)	1909.2 (439.6)	1754.5 (344.5)	2176.1 (458.5)	2.2 × 10^−16^
Potatoes (g/day)	86.2 (39.1)	82.8 (37.1)	91.9 (41.8)	0.00062
Potatoes (RM, continuous)	6.0 (1.4)	6.0 (1.5)	6.1 (1.3)	0.13
Q1		69 (≤5.88)	40 (≤5.85)	
Q2		69 (5.89–6.36)	40 (5.86–6.46)	
Q3		69 (6.40–6.70)	40 (6.46–6.94)	
Q4		69 (>6.71)	40 (>6.94)	
Olive oil (g/day)	32.7 (10.3)	33.3 (9.9)	31.8 (11.0)	0.14
Olive oil (RM)	4.8 (1.0)	4.9 (0.9)	4.8 (1.1)	0.33
Alcoholic beverages (g/day)	73.1 (156.1)	26.3 (78.3)	154.6 (214.1)	2.2 × 10^−16^
Alcoholic beverages (RM)	1.2 (4.9)	−0.7 (4.1)	4.4 (4.4)	2.2 × 10^−16^
Non-consumers		191 (≤−3.23)	36 (≤−1.89)	
Below-median consumers		42 (2.72–4.91)	63 (−0.57–6.85)	
Above-median consumers		43 (>4.96)	61 (>6.88)	
Blue fish (g/day)	16.7 (8.0)	16.4 (8.0)	17.2 (7.9)	1.3 × 10^−8^
Blue fish (RM)	3.7 (1.4)	3.7 (1.3)	3.8 (1.5)	0.0043
Q1		69 (≤3.86)	40 (≤3.86)	
Q2		69 (3.87–4.01)	40 (3.87–4.12)	
Q3		69 (4.02–4.15)	40 (4.13–4.30)	
Q4		69 (>4.15)	40 (>4.30)	
Caloric beverages (g/day)	32.2 (97.1)	18.8 (69.3)	55.6 (129.2)	0.00017
Caloric beverages (RM)	−0.8 (4.2)	−1.4 (3.7)	0.3 (4.8)	0.11

BMI: body mass index; RM: residual method. * Wilcoxon rank-sum test for continuous variables and Pearson chi-square test for categorical variables.

**Table 2 nutrients-16-03131-t002:** Number of differentially expressed genes in the top 5 food groups adjusted by the residual method (FDR < 0.05).

	Top 5 Food Groups				
	Potatoes	Caloric Beverages	Olive Oil	Blue Fish	Alcoholic Beverages
**DEG**	n	n	n	n	n
Upregulated	1242	1093	618	214	186
Downregulated	1344	693	369	324	216
Total	2586	1786	987	538	402
**DEG adjusted by food groups (sensitivity analysis)**					
Upregulated	142	5	2	-	-
Downregulated	84	5	3	-	-
Total	226	10	5	-	-

DEG: differentially expressed genes; FDR false discovery rate.

**Table 3 nutrients-16-03131-t003:** Summary of key findings from functional protein–protein interaction (PPI) network analyses of the 3 main food groups.

Food Group	Reactome ID	Description	GeneRatio	BgRatio	*p*-Value	Genes
Blue fish	R-HSA-69620	Cell cycle checkpoints	30/260	293/10554	2.89 × 10^−11^	*CENPE/ERCC6L/SGO2/CENPF/CLSPN/KNL1/SGO1/DBF4/BRCA1/MCM6/RMI1/SKA2/PSMD12/BARD1/PPP2R5E/ORC5/PSME3/BRCC3/KIF2A/SEM1/PSME4/RAD50/RANBP2/YWHAB/CKAP5/MAPRE1/SEH1L/AHCTF1/NUP160/COP1*
	R-HSA-2470946	Cohesin loading onto Chromatin	6/260	10/10554	4.08 × 10^−8^	*MAU2/SMC3/RAD21/PDS5B/NIPBL/PDS5A*
	R-HSA-6811442	Intra-Golgi and retrograde Golgi-to-ER traffic	19/260	200/10554	4.72 × 10^−7^	*CENPE/KIF11/KIF20B/KIF15/PLA2G6/GOLGA4/KIF21A/KIF2A/KLC4/KIF5B/RINT1/TRIP11/GOLGA5/GCC2/TMF1/ARFGAP2/GOSR1/MAN1A2/DCTN1*
	R-HSA-72203	Processing of capped Intron-containing pre-mRNA	19/260	243/10554	8.68 × 10^−6^	*SNRPF/SNRPD1/SMNDC1/THOC7/RANBP2/CWC22/THOC5/CSTF3/PRPF40A/DHX16/CPSF2/SUGP1/SEH1L/CD2BP2/POLR2D/NUP160/NUP153/HNRNPU/RBM17*
	R-HSA-199992	Trans-Golgi network vesicle budding	10/260	72/10554	9.68 × 10^−6^	*TFRC/ARRB1/AP1G2/FTL/CPD/AP4M1/CLTC/AP3B1/PIK3C2A/GOLGB1*
	R-HSA-3371556	Cellular response to heat stress	11/260	88/10554	9.81 × 10^−6^	*HSPA4L/HSPA5/HSPH1/PTGES3/DNAJC2/RANBP2/HSPA4/HDAC6/SEH1L/NUP160/NUP153*
	R-HSA-5693568	Resolution of D-loop structures through Holliday junction intermediates	7/260	33/10554	1.25 × 10^−5^	*BRCA2/BRCA1/RMI1/BARD1/RAD50/PALB2/SPIDR*
	R-HSA-983169	Class I MHC mediated antigen processing and presentation	24/260	371/10554	1.48 × 10^−5^	*HSPA5/PDIA3/PSMD12/KLHL21/LRSAM1/UBE2K/SEC24A/KCTD6/PSME3/FBXO31/ASB13/SEM1/RNF123/PSME4/TRIM11/IKBKB/ATG7/RNF220/CUL2/FBXO11/UBA5/UBE3A/CUL1/TRIP12*
	R-HSA-5663202	Diseases of signal transduction	23/260	378/10554	5.85 × 10^−5^	*HDAC5/PSMD12/ARRB1/ERBB3/ERBB2/ARRB2/PPP2R5E/PSME3/KRAS/SEM1/PSME4/YWHAB/TRIM24/ADAM10/ATG7/PTPN11/HDAC6/HDAC2/AP3B1/POLR2D/APC/CUL1/RASA1*
Alcoholic beverages	R-HSA-163200	Respiratory electron transport, ATP synthesis by chemiosmotic coupling, and heat production by uncoupling proteins.	13/177	123/10554	1.32 × 10^−7^	*NDUFB6/COX6C/NDUFC1/NDUFAB1/NDUFA1/UQCR11/NDUFAF4/ATP5PD/UQCRB/ATP5F1E/NDUFS6/UQCRQ/NDUFB5*
	R-HSA-983695	Antigen activates B cell receptor (BCR), leading to generation of second messengers	7/177	32/10554	7.83 × 10^−7^	*CD22/CD19/CD79B/BLK/CD79A/PLCG2/PTPN6*
	R-HSA-5368286	Mitochondrial translation initiation	9/177	87/10554	1.43 × 10^−5^	*MRPL33/MRPS36/MRPL13/MRPL15/CHCHD1/MRPS35/MRPS22/MTFMT/MRPL19*
	R-HSA-6811442	Intra-Golgi and retrograde Golgi-to-ER traffic	11/177	200/10554	5.40 × 10^−4^	*CYTH2/GOLGA4/ACTR10/BET1L/MAN2A1/DCTN1/VPS54/STX5/CYTH1/TMF1/VPS52*
Potatoes *	R-HSA-72203	Processing of capped Intron-containing pre-mRNA	11/131	243/10554	2.14 × 10^−4^	*POLR2L/LSM7/AAAS/PPIL4/GTF2F2/THOC5/POLDIP3/RANBP2/DHX38/PRPF40A/EFTUD2*
	R-HSA-9020558	Interleukin-2 signaling	3/131	12/10554	3.79 × 10^−4^	*IL2RG/PTK2B/SHC1*
	R-HSA-381038	XBP1(S) activates chaperone genes	5/131	57/10554	6.81 × 10^−4^	*CTDSP2/PREB/DCTN1/SHC1/GOSR2*
	R-HSA-446203	Asparagine *N*-linked glycosylation	10/131	302/10554	4.33 × 10^−3^	*UGGT2/MPDU1/PREB/RNF5/ARFGAP2/GMPPA/COG4/B4GALT3/DCTN1/GOSR2*
	R-HSA-3108232	SUMO E3 ligases SUMOylate target proteins	7/131	181/10554	7.29 × 10^−3^	*HIC1/AAAS/SMC6/SMC3/RANBP2/DAXX/TOPORS*

* Adjusted by correlated food variables.

## Data Availability

The datasets generated and analyzed in our study are available upon reasonable request from the corresponding authors (v.moreno@iconcologia.net; mireiaobon@iconcologia.net).

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
