# Peer review of "Diet Impacts on Gene Expression in Healthy Colon Tissue: Insights from the BarcUVa-Seq Study"

_nutrients, 2024, doi:10.3390/nu16183131_

Round 1

Reviewer 1 Report

Comments and Suggestions for Authors

This interesting study evaluated the impact of diet on samples of healthy colon tissue, drawing from a previous cohort of patients from an earlier study that aimed to screen for colorectal cancer in the general Spanish population. The sample size is significant, with over 400 patients included. The methodology is very clear, detailed, and relevant.

Minor considerations:

1) There is an extra parenthesis on line 57;

2) In the introduction, I would specify that, compared to the general population, patients with colon diseases, particularly IBD, also show a reduced propensity for regular physical activity (https://pubmed.ncbi.nlm.nih.gov/38077160/), which predictably impacts the genetic lines analysed by the authors.

Author Response

We thank the reviewer for his/her careful reviewing of the manuscript and positive comments.

Comment 1) There is an extra parenthesis in line 57.

Response 1) After carefully reading the line, we found that this parenthesis came from the previous line and might be confounded by the acronym presentation of the mediterranean diet pattern: “patterns associated with higher intake of plant-based foods with regular intake of fish and seafood (such as the Mediterranean Diet (MedDiet) or the Prudent pattern) are associated with a decreased colorectal cancer risk”.

Comment 2) In the introduction, I would specify that, compared to the general population, patients with colon diseases, particularly IBD, also show a reduced propensity for regular physical activity (https://pubmed.ncbi.nlm.nih.gov/38077160/), which predictably impacts the genetic lines analyzed by the authors.

Response 2) We thank the reviewer for highlighting this very interesting topic. Indeed, physical activity, or physical inactivity, affects both CRC and inflammatory bowel diseases. In fact, we already emphasize this in the first paragraph of the introduction “Despite having different clinical manifestations and severities, these diseases share several risk factors, particularly those related to lifestyle, including obesity, tobacco smoking, and sedentary behavior”.

However, as the reviewer mentioned, we have cited the article by Antonietta Gerarda Gravina et al., as it specifically addresses IBD, along with the article by Carreras-Torres et al. (Reference #5)

Reviewer 2 Report

Comments and Suggestions for Authors

The authors aimed to evaluate the biological mechanisms, specifically mRNA levels, through which diet, including specific foods, may influence gastrointestinal diseases. Their findings indicate that food intake affects gene expression in the colon of individuals without tumor pathology. Notably, the consumption of blue fish, alcohol, and potatoes was associated with the regulation of genes involved in pathways such as DNA repair, immune system function, and protein glycosylation.

The subject of the manuscript is both interesting and relevant to the scope of the journal, addressing an important issue. The authors' findings and observations are compelling, and I recommend the publication of the manuscript after minor revisions.

The introduction is somewhat repetitive in discussing the effects of nutrition on the prevention or induction of gastrointestinal diseases. In my opinion, a more detailed explanation of the potential mechanisms that could prevent or induce these diseases should be included.

I understand this may be outside the scope of the current work, but since the average BMI is in the overweight range, a brief discussion of its relevance in the discussion section would be valuable. What is the relationship between BMI and gastrointestinal diseases? Is there any variation in gene expression when comparing volunteers with normal BMI to those who are overweight?

Regarding line 331, it's important to note that alcohol metabolism is mediated by CYP enzymes primarily when concentrations are high. Regular alcohol consumption, such as a glass of wine, is mainly metabolized by alcohol dehydrogenase.

Another crucial point is the way food is cooked or processed, which should be commented on and discussed. For example, fried potatoes likely have a different impact on gene expression compared to boiled potatoes, and the same applies to other foods. The balance between saturated fatty acids and omega-3 fatty acids differs in oil-fried fish compared to grilled fish.

Author Response

We thank the reviewer for his/her careful reviewing of the manuscript and positive comments.

Comment 1) The introduction is somewhat repetitive in discussing the effects of nutrition on the prevention or induction of gastrointestinal diseases. In my opinion, a more detailed explanation of the potential mechanisms that could prevent or induce these diseases should be included.

Response 1) We thank the reviewer for his/her insightful thoughts. We agree that the introduction could be more detailed regarding the potential mechanisms that could prevent or induce bowel diseases. This is now reflected in the introduction section, lines (74-84).

Comment 2) I understand this may be outside the scope of the current work, but since the average BMI is in the overweight range, a brief discussion of its relevance in the discussion section would be valuable. What is the relationship between BMI and gastrointestinal diseases? Is there any variation in gene expression when comparing volunteers with normal BMI to those who are overweight?

Response 2) We thank the reviewer for this suggestion, which we think will improve the relevance of our study. We have added a brief discussion (lines 200-203; 259-263). In addition, we have added a Supplementary Table 2 in which the five main food groups are analyzed after adding BMI as another adjustment variable to measure the impact of the BMI in our analysis. Overall, at least 65% of the statistically significant DEA genes are retained when adjusting by BMI on top five food groups. LogFC spearman’s rho correlation values ranged between 0.993 and 0.998 when comparing top 5 food groups with and without BMI adjustment. Regarding the variation in gene expression when comparing volunteer with normal BMI to those who are overweight, we would like to highlight that no great differences are observed. However, this is a matter covered in a related work which is now also under submission.

Comment 3) Regarding line 331, it's important to note that alcohol metabolism is mediated by CYP enzymes primarily when concentrations are high. Regular alcohol consumption, such as a glass of wine, is mainly metabolized by alcohol dehydrogenase.

Response 3) We thank the reviewer for this comment, as it has allowed us to be more precise in the discussion. See the modification on lines 352-354.

“This occurs because alcohol metabolism, primarily mediated by alcohol dehydrogenase and aldehyde dehydrogenases at low alcohol concentrations, and by catalase and cytochrome P450s (CYP2E1) at high concentrations, , results in the generation of derivate compounds, including reactive oxygen species (ROS) and decreases the levels of glutathione”

. In addition, we have carefully checked how the genes conforming alcohol dehydrogenase (ADH) and CYP450 complexes are expressed in our samples, and none of them are statistically significant when comparing alcohol consumption. This could be explained because alcohol is mainly metabolized by ADH and/or CYP450 in hepatocytes rather than in the mucosal gut, therefore indirect alcohol consumption effects are seen.

Comment 4) Another crucial point is the way food is cooked or processed, which should be commented on and discussed. For example, fried potatoes likely have a different impact on gene expression compared to boiled potatoes, and the same applies to other foods. The balance between saturated fatty acids and omega-3 fatty acids differs in oil-fried fish compared to grilled fish.

Response 4)

We completely agree with the reviewer that the cooking method affects the composition of foods and, consequently, may impact gene expression. We raised this issue regarding potatoes consumption in the discussion section (lines 398-402), Nevertheless, in the methodology section (Lines 173-174), we acknowledge that the variable ‘potatoes’ we used includes both cooked and fried potatoes. We were unable to present the results separately for statistical reasons.

In our population, olive oil is the preferred cooking oil. We observed a strong correlation between these two variables (potatoes and olive oil). By presenting the results for potatoes adjusted for olive oil, we aimed to mitigate the differences between fried and non-fried potatoes. We have focused only on mentioning and discussing the gene expression of these results. Also, it should be noted that we have analyzed different ways of red meat consumption (red meat, cured meat and processed meat). Results are shown on Supplementary Table 1.

Following the reviewer suggestion, we have added to the limitations section that we could not account for the cooking method for all the variables used, and therefore, studies with this type of information could be more accurate. See lines 398-402

“Finally, it should be noted that previous studies, including the one we currently present, have not been able to account for the cooking method for all variables, which may affect gene expression. More studies with this level of detail are required to obtain more accurate results.”

Reviewer 3 Report

Comments and Suggestions for Authors

Reviewer comments

In this paper, the Synergistic Diet impacts on gene expression in healthy colon tissue: Insights from the BarcUVa-Seq Study. In my opinion, this paper could be accepted after considering the following major issues.

1. Improve the abstract section and highlighted the aim of this study.

2. In the introduction, before starting with the objectives of this work, please summarise why this work is needed.

3. Line 53-55. Write disease risk factor two or three examples

4. Line 62 -64. need to rewrite

5. Line 67 to 69.  “Peroxisome proliferator-activated receptor (PPAR) are a family of ligand transcription factors that regulate the expression of genes participating in cell differentiation and lipid metabolism” to “Peroxisome proliferator-activated receptors (PPARs) are a group of nuclear receptor proteins that act as transcription factors that regulate the expression of genes participating in cell differentiation and lipid metabolism”.

5. Line 74. Please write the hypothesis

7. Line 369-375 need revise it.

6. Please discuss and updated the whole manuscript with recent references.

5. Please check the grammar and sentence usage patterns in the result and discussion part.

6. A few type errors can be found in this manuscript.

Author Response

We thank the reviewer for his/her careful reviewing of the manuscript and positive comments.

Comment 1) Improve the abstract section and highlighted the aim of this study.

Response 1) Changes are highlighted in red in the abstract section (Lines 23-38).

Comment 2) In the introduction, before starting with the objectives of this work, please summarise why this work is needed.

Response 2) We have highlighted the rational in the introduction. Lines (62-65).

Comment 3) Line 53-55. Write disease risk factor two or three examples

Response 3) Lines 55-61 on the original manuscript contain three examples on gastrointestinal disease risk factors: “patterns associated with higher intake of plant-based foods with regular intake of fish and seafood (such as the Mediterranean Diet (MedDiet) or the Prudent pattern) are associated with a decreased colorectal cancer risk, while those characterized by higher intake of red or processed meats, sweets and high saturated fat products (such as the Western pattern) are associated with an increased risk [7]. Similarly, a high-quality diet index is linked to a reduced risk of inflammatory bowel disease”

Comment 4) Line 62 -64. need to rewrite

Response 4) We have slightly modified this sentence to make it clearer.

Comment 5) Line 67 to 69.  “Peroxisome proliferator-activated receptor (PPAR) are a family of ligand transcription factors that regulate the expression of genes participating in cell differentiation and lipid metabolism” to “Peroxisome proliferator-activated receptors (PPARs) are a group of nuclear receptor proteins that act as transcription factors that regulate the expression of genes participating in cell differentiation and lipid metabolism”.

Response 5) Thanks for this suggestion. It is now modified in lines 67-69.

Comment 6) Line 74. Please write the hypothesis

Response 6)  As stated in line 87: ”The aim of the present study was to assess and replicate, in a population of healthy Spanish adults, the associations between dietary components and patterns and gene expression in colon tissue, and therefore, elucidate biological mechanisms by which diet may influence on bowel diseases.”

Comment 7) Line 369-375 need revise it.

Response 7) Lines 369-375 (now Lines 389-392) have been revised. Some modifications have been made to make the statements clearer and are highlighted in red.

Comment 8) Please discuss and updated the whole manuscript with recent references.

Response 8) Changes are highlighted in red, also in reference section., which have been updated and extended.

Comment 9) Please check the grammar and sentence usage patterns in the result and discussion part.

Response 9) Grammar and sentence usage in results and discussion checked. Changes are highlighted in red.

Comment 10) A few type errors can be found in this manuscript.

Response 10) We have double-checked all the manuscript and corrected all the typos that we have found (changes are tracked and red coloured).